# Recent Advancement in Assessment and Control of Structures under Multi-Hazard

Matin Jami [1], Rajesh Rupakhety [1,*], Said Elias [2], Bjarni Bessason [3] and Jonas Th. Snæbjörnsson [4]

1 Earthquake Engineering Research Centre, Faculty of Civil and Environmental Engineering, School of Engineering and Natural Science, University of Iceland, Austurvegur 2a, 800 Selfoss, Iceland; amj38@hi.is

2 Department of Construction Management and Engineering (CME), Faculty of Engineering Technology (ET), University of Twente (UTWENTE), 7522 NB Enschede, The Netherlands; elias.rahimi@utwente.nl

3 Faculty of Civil and Environmental Engineering, School of Engineering and Natural Science, University of Iceland, 102 Reykjavík, Iceland; bb@hi.is

4 Department of Engineering, School of Technology, Reykjavik University, 101 Reykjavík, Iceland; jonasthor@ru.is

* Correspondence: rajesh@hi.is; Tel.: +354-5254129

**Abstract:** This review presents an up-to-date account of research in multi-hazard assessment and vibration control of engineering structures. A general discussion of the importance of multi-hazard consideration in structural engineering, as well as recent advances in this area, is presented as a background. In terms of performance assessment and vibration control, various hazards are considered with an emphasis on seismic and wind loads. Although multi-hazard problems in civil engineering structures are generally discussed to some extent, the emphasis is placed on buildings, bridges, and wind turbine towers. The scientific literature in this area is vast with rapidly growing innovations. The literature is, therefore, classified by the structure type, and then, subsequently, by the hazard. Main contributions and conclusions from the reported studies are presented in summarized tables intended to provide readers with a quick reference and convenient navigation to related publications for further research. Finally, a summary of the literature review is provided with some insights on knowledge gaps and research needs.

**Keywords:** multi-hazard; earthquake; wind; flood; hazards; hurricane; mitigation; resilience; risk assessment; bridge; building; wind turbine; control system

## 1. Introduction

Natural hazards, such as earthquakes and wind forces, pose a challenge for human safety and comfort. Forces generated by these natural processes can damage, or even collapse, vulnerable civil engineering structures. The risk to lives and properties posed by natural hazards increases with urbanization, where large cities and metropolitan areas get more and more densely populated. Increasing urbanization and shortage of land results in the need to build taller and more complex structures which can be more vulnerable to lateral forces created by wind and earthquakes. Effects of natural hazards on civil engineering structures is, therefore, an important field of research.

Between 1998 and 2017, natural disasters affected 4.4 billion people worldwide, caused 1.3 million casualties [1], and resulted in economic loss of 2900 billion USD. During this 20-year period, floods, storms, and earthquakes were the most frequent hazards, accounting for 43.4, 28.2, and 7.8% of all natural disasters, respectively. Although floods were the most frequent hazard during this time, earthquakes and storms have been the deadliest and the costliest, respectively. Floods and earthquakes, combined, killed nearly one million people and resulted in an economic loss of almost 2000 billion USD during this 20-year period. The frequencies, casualties, and economic losses, caused by different types of natural hazards

between 1998 and 2017, are shown in Figure 1. The numbers in Figure 1, which are based on CRED report [1], clearly show that earthquakes and storms are the most damaging natural hazards. It is interesting to note that earthquakes have killed more people than all other hazards combined.

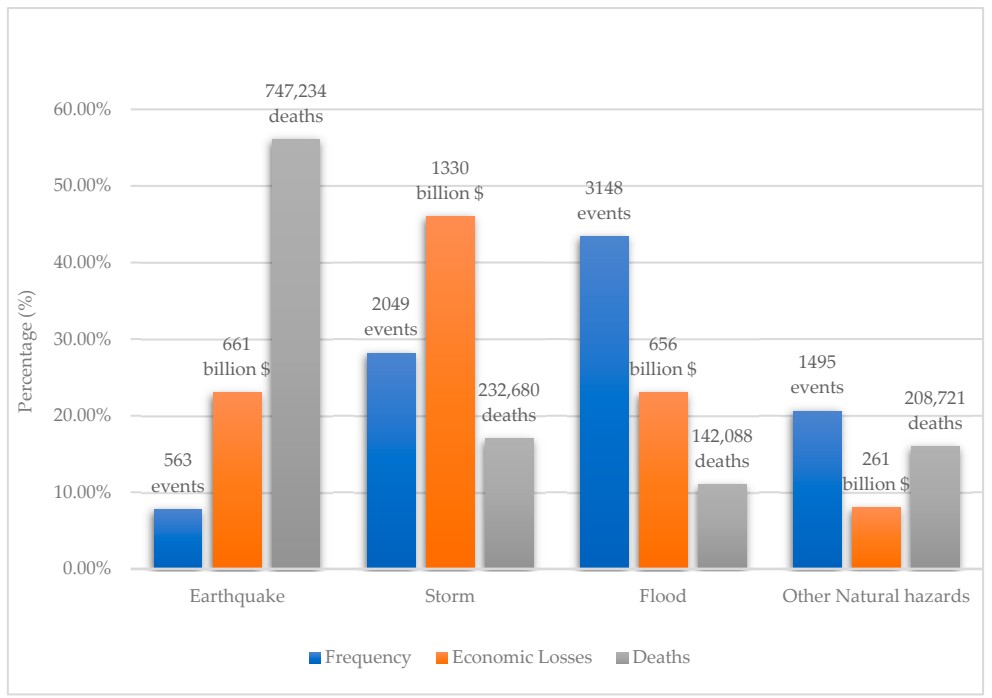

**Figure 1.** Frequencies of different natural hazards and their effects during 1998–2017 (based on CRED report, [1]).

Different natural processes affect structures and people in different ways. While the simultaneous occurrence of two different types of damaging hazards, such as strong wind and earthquake, is rare, some natural processes can induce secondary hazards. For example, fire and landslides are known to occur after strong earthquakes (see, for example, Ravankah et al. [2]). Moreover, a structure may be exposed to different types of natural hazards, albeit not simultaneously, during its lifetime. Therefore, it needs to be resistant to forces and damage mechanisms imposed by more than one natural process. Structures optimally designed for actions from one type of natural hazard may not necessarily be well equipped to deal with actions from all types of hazards. This leads to the need for hazard mapping, considering different types of natural processes and their interdependencies.

Consideration of multiple hazards in urban development is gaining popularity in the research community. For example, Bathrellos et al. [3] studied probabilities of incidence of floods, landslides, and earthquakes, in a specific area in Northeastern Greece, to map multiple hazards and identify areas suitable for urban development. Hicks et al. [4] explore disaster risk reduction from a multi-hazard perspective. Regional multi-hazard mapping for urban development is gaining popularity in research (e.g., [5]). Vulnerability and design of structures against multiple hazards is also gaining popularity in research. As an example, Aly [6], as well as Aly and Abburu [7], discuss some fundamental differences between wind and earthquake-resistant designs of high-rise buildings. A review of studies on the vulnerability of buildings subjected to wind and earthquake forces is presented by Indirli et al. [8]. A framework for life-cycle loss estimation in tall buildings subjected to wind and seismic forces is presented by Venanzi et al. [9]. Civil engineering infrastructure, such as dams, bridges, roads etc., are lifelines of modern society. Although multi-hazard risk assessment of infrastructure is challenging [10,11], it is an important tool to improve their safety and operability following natural disasters, which is instrumental for social resilience. Various factors affecting costs and performance of infrastructure in a multi-hazard environment is

discussed in Ettouney and Alampalli [12]. Performance and fragilities of special structures, such as dams and floodwalls exposed to multiple hazards, are studied in Ardebili and Saouma [13] and Bodda [14].

Natural events, such as wind and earthquakes, impose dynamic forces on buildings and other civil engineering structures. Damage caused by such forces depends on the dynamic properties of the structure, as well as the characteristics of the wind forces and ground motion. In most cases, structural damage is a result of excessive vibration. Vibration control, which refers to reducing oscillations of structures exposed to dynamic forces, can, therefore, be used as a protective measure. Vibration control makes use of active, passive, or hybrid secondary devices that are installed on the structure and designed/tuned/actuated for optimal reduction in structural responses such as displacement, acceleration, etc. Base-isolation, for example, has been a popular and effective protection against earthquakes (see, for example, [15–19]). Tuned mass dampers (TMD) and other supplemental damping devices of different designs and configurations have also been known to effectively reduce wind and earthquake-induced vibrations of different types of structures (see, for example, [20–22]). Vibration control systems can provide an alternative protection for existing structures where retrofitting or strengthening is considered too costly or not feasible, due to factors such as aesthetics, cultural aspects, etc. Control devices that are effective against the forces generated by one type of natural hazard might not be effective against other hazards. For example, base isolation systems, which are effective for seismic protection of structures, might cause an adverse response during strong wind [23]. Due to the uncertainties in the amplitude and frequency content of dynamic forces, induced by wind and earthquakes, and their relationship with the properties of the affected structures, consideration of a multi-hazard scenario is especially important when designing vibration control systems.

This work is an attempt to bring together and synthesize valuable information and conclusions presented in a vast body of research literature on multi-hazard effects and control of civil engineering structures. The work is based on a review and synthesis of published literature. Relevant studies were searched through scholarly databases such as Web of Science, Google Scholar, and Scopus. The keywords used for searching were "multi-hazard", "vibration-control", "seismic control", "tuned mass dampers", "seismic fragility", and "life-cycle assessment". The search results were then narrowed first by scanning the titles of articles to include only those that indicated relevance in the multi-hazard problem, addressing one or more of the criteria: (a) hazard mapping/quantification, (b) performance assessment, (c) design and/or optimization, (c) fragility assessment, (d) life-cycle and/or cost-benefit analysis, and (e) vibration control. This resulted in more than 400 articles. The Abstract and Conclusion sections of these articles were then studied to further filter out studies that did not address the multi-hazard problem. This resulted in 220 articles. The references listed in these articles were then checked to search for more relevant articles. Special attention was given to state-of-the art review studies. References listed in studies were checked in detail to search for additional relevant articles. In total, 263 articles were studied and are referenced to in this work. Among these, there are 210 journal articles, 17 books/reports, 11 theses, 14 book chapters, and 11 conference papers. These include 18 state-of-the art reviews.

Initial thematic development of the work was, first and foremost, based on the keywords listed in these articles. The keywords in these articles were extracted, and their frequencies were counted. In total, 699 unique keywords were found. Multi-word keywords were then replaced by a single word (called, here, a reduced keyword) that is representative of the scope of the work. For example, "vibration control" was reduced to "control", "risk assessment" was reduced to "risk", and so on. In some cases, such as "wind turbines", both words were retained. This resulted in 117 keywords. Similar keywords were then grouped together to identify themes/scopes. For example, "seismic", "ground motion", and "earthquakes" were placed under the theme of "Seismic". The number of occurrences of these themes were then counted, and the themes were ranked. Frequency distribution of the most frequent themes is presented in Figure 2. Some reduced keywords

with a low frequency of occurrence are therefore not considered useful in creating an overall theme of the subject being studied and are not shown in the figure. Control is the most frequent theme, and multi-hazard is the third most frequent theme. Seismic and wind loads are the most frequently considered hazards. In terms of structures, bridges, buildings, and wind turbines are frequent themes, while only a few (less than 10) occurrences of other infrastructure and lifelines were encountered. This thematic distribution of the studied articles was used to prepare the main structure of this paper, which is schematically presented in Figure 3.

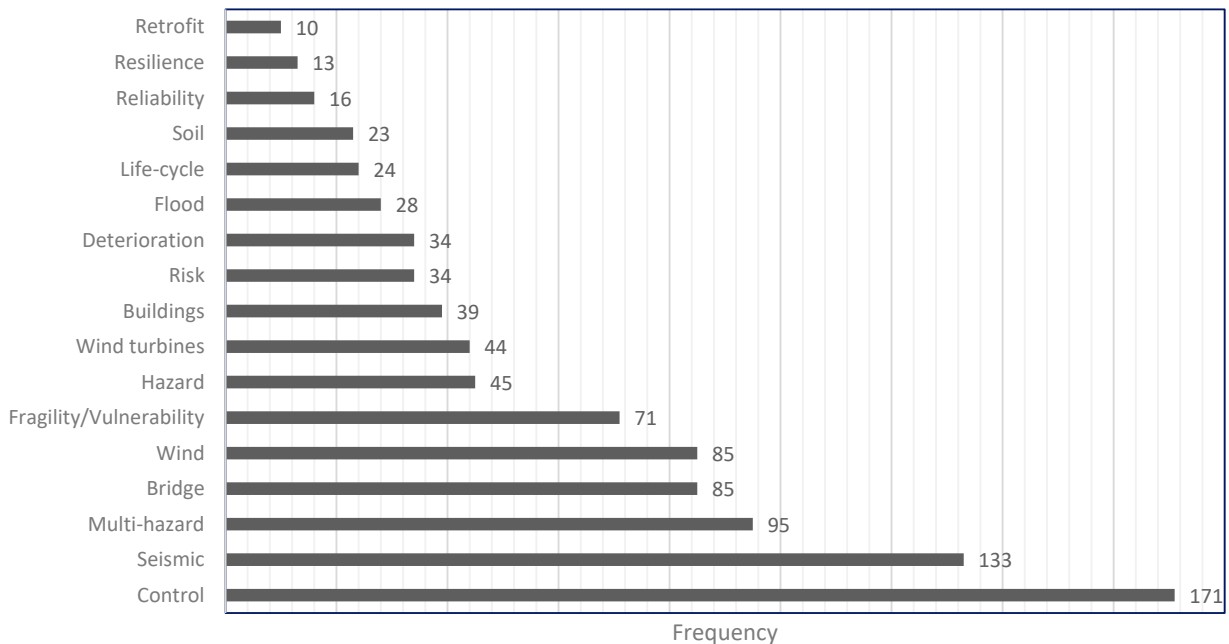

**Figure 2.** Frequency distribution of the most relevant themes extracted from reduced keywords.

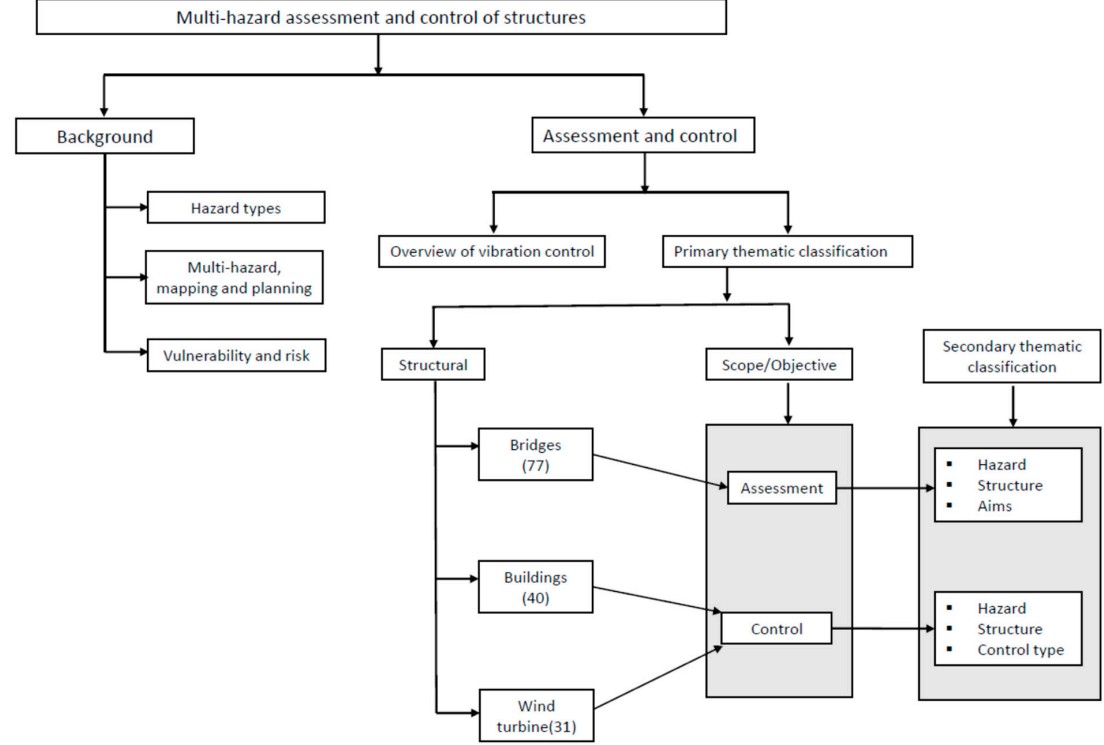

**Figure 3.** Schematic representation of thematic classification and organization of the paper.

Background information about different types of hazards, multi-hazard scenarios, and associated vulnerability and risk is provided in Section 2. This section is not a state-of-the-art review of these topics but rather background information for the rest of the paper (see Figure 3). The main part of the paper, which is the state-of-the-art review part of the paper, is briefly termed as assessment and control (see Figure 3). The review is based on the themes encountered in the studied literature. Topics such as fragility/vulnerability assessment, life-cycle assessment, multi-hazard assessment, and reliability assessment are covered under the assessment theme, while the control theme mainly deals with vibration suppression. As vibration control is the most dominant theme of the studied papers (see Figure 2), a brief literature review of different control systems is provided in Section 3. The review is primarily classified by two themes: namely, structure and scope of work. Bridges, buildings, and wind turbines are covered in Sections 4–6, respectively. Each of these sections is sub-divided into assessment and control sub-sections. The literature on bridges is dominated by the assessment theme, which is classified into secondary themes such as hazard type, type of bridge, and the main aims of the study. The literature on buildings and wind turbines contains several studies of multi-hazard vibration control. For each of these structures, the studies reviewed here are sub-classified into secondary themes of hazard, type of building/wind turbine, and the type of control device.

## 2. Risk: Hazard, Exposure, and Vulnerability

Risk related to disasters (disaster risks) can include loss of lives, disrupted economy, damages to the environment, etc. Risk is linked to the combination of hazard, physical exposure, and vulnerability of the infrastructure. The roles of each of these factors are briefly reviewed in the following sections.

### 2.1. Hazard

The definition of "hazard" in a broader sense is "any external or internal process or event that might degrade the performance of the system on hand" [12]. The United Nations General Assembly [24] defines hazard as "a process, phenomenon or human activity that may cause loss of life, injury or other health impacts, property damage, social and economic disruption or environmental degradation".

Natural events, such as storms, earthquakes, or floods, are well-known hazards with widespread potential to turn into disasters. While these events are mostly sudden and occur in a relatively short time window, slower processes, such as fatigue, corrosion, ageing, etc., can also impact structural performance over their life span.

Among the three elements that constitute disaster risk, hazard is the one that is mostly beyond human control. Nevertheless, a proper understanding of the occurrence frequency, spatio-temporal distribution, and intensity of the hazard is important for disaster risk reduction. Recent advances in sensing technology: data collection, processing, storage, sharing capabilities; and modelling/computational tools have improved our understanding how different hazards affect civil engineering structures. Hazards can be of different types. For example, they can be natural events, such as earthquakes, or man-made ones, such as explosions.

Different classifications of hazard have been proposed for multi-hazard studies. For example, Ettouney and Alampalli [12] discuss the classification of hazards based on Temporal, Frequency, and Newtonian characteristics. Temporal characterization distinguishes between simultaneous occurrence, segregation in time, and cascading effects. Frequency characterization distinguishes continuous processes, such as corrosion, from intermittent processes such as earthquakes. Intermittent processes can be further classified as frequent, intermediate, or rare. Newtonian characterization is another useful approach for hazard classification that is generally used in design codes. In design codes, hazards are generally quantified in terms of loads, such as wind load, earthquake load, etc. The impact of these loads can be quantified by different metrics, such as stress, deformation, etc., and are evaluated based on Newtonian mechanics. Such hazards have been termed as Newto-

nian [12]. Other processes, such as corrosion, wear and tear, fatigue, etc., are termed as non-Newtonian [12]. Natural hazards can also be classified based on their origin and the geo-atmospheric processes associated with them, such as

1. Biophysical (wildfire).
2. Atmospheric (wind or thunderstorms, lightning, hail, snow, and climate change).
3. Hydrological (drought and flood).
4. Shallow Earth Processes (erosion, subsidence and uplift, and mass movement).
5. Geophysical (volcanic eruption, tsunami, landslide, earthquake, and snow avalanche).

The idea that a structure needs to resist different types of hazards during its service life is well-established in civil engineering. For example, design codes and standards have provisions for different types of actions such as dead load, live load, wind load, seismic load, etc. Simultaneous occurrence of multiple actions is addressed in design codes through load combinations. Such recognition of multiple actions and load combinations does not encompass the real extent of multi-hazard effects and interactions. Multi-hazard generally refers to the concept where two or more hazards interact through structural performance. A multi-hazard interaction, for example, can impact risk due to a hazard when a decision regarding structural exposure and vulnerability against frequency, location, and amplitude of another hazard is made. For example, a change in design wind load and/or structural capacity can impact structural vulnerability to earthquake forces. Multi-hazard interactions may result in common or conflicting design solutions. For example, provision of structural ductility is beneficial for both blast and seismic loads.

Padgett and Kameshwar [25] present a comprehensive classification of multi-hazard combinations for bridges. Although their classification was intended for bridges, it can be generalized for most civil engineering structures, as is presented in Figure 4. Classification of hazard, according to Figure 4, helps to understand potential interactions between different hazards through their effects on structures.

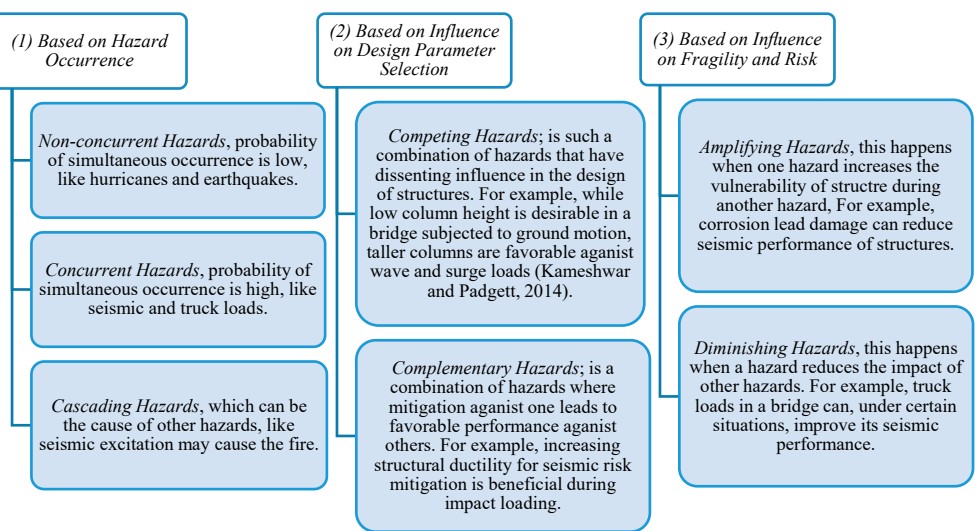

**Figure 4.** Classification of multi-hazard combinations [26] (modified from Padgett and Kameshwar [25]).

Multi-hazard consideration is important for structural safety and reliability. Duthinh and Simiu [27] present an interesting point regarding the traditional practice of treating different hazards independently and designing structural components based on the more demanding hazards. Taking an example of wind and earthquakes, they show that the ASCE Standard 7 provisions are not risk-consistent in the sense that, in regions affected by both strong wind and earthquakes, risks of exceedance of limit states can be up to twice as high as those in regions where only one of these hazards dominates. Kappes et al. [28] discuss the challenges of analyzing multi-hazard risk and existing frameworks to address those challenges. Zaghi et al. [29] presents the limitations of modern design codes in adequately

addressing multi-hazard risk and emphasizing the need for common nomenclature for multi-hazard design. They also mention several problems and challenges in the multi-hazard design of structures. Different aspects of multi-hazard approaches, to mitigate risk to civil engineering infrastructure, are further discussed in Gardoni and Lafave [30]. The implications of considering potential multi-hazard effects in the life cycle cost analysis of an infrastructure is addressed by Jalayer et al. [31] and Fereshtehnejad and Shafieezadeh [32].

### 2.2. Exposure, Multi-Hazard Mapping and Planning

In the context of disaster risk, the UNDRR (United Nations Office for Disaster Risk Reduction) defines exposure as "the situation of people, infrastructure, housing, production capacities and other tangible human assets located in hazard-prone areas" [33]. Exposure is a necessary factor for disaster risk. Exposure is one of the risk determinants that can be controlled, to some extent, by proper planning. Such decisions are, however, not feasible in cases of existing risk: for example, large cities already built-in hazardous space. Reducing exposure to multi-hazards is more challenging than if only a single hazard is considered. Urban planning, land-use policy-making, environmental protection decisions, etc., need to rely on, and benefit from, multi-hazard considerations.

Local and regional scale mapping of different types of hazards is essential in multi-hazard considerations when analyzing exposure. Although significant advancements have been made in mapping individual hazards, mapping multi-hazard is challenging due to the differences in their physical phenomena, measures of frequency/amplitude, impact on structures, etc.

Barua et al. [34] present a multi-hazard map for different districts of Bangladesh based on local historical disaster database and comparison of scenario hazard scales with those in other countries. Their study includes earthquakes, tornadoes, floods, and cyclones, which are combined through a weighing scheme. Pourghasemi et al. [35] present multi-hazard mapping of Fars Province in southern Iran. They consider floods, fires, and landslides. They test two different machine learning algorithms in predicting distribution of these hazards based on historical data, and make use of different conditioning factors such as aspect, elevation, drainage, annual mean rainfall, etc. They highlight the importance of multi-hazard mapping in land-use planning, sustainable development, and watershed management in the study region. A similar study for the western region of Iran is presented in Pourghasemi et al. [36].

For multi-hazard mapping of relatively small areas, the Analytical Hierarchy Process (AHP) has been proposed as a suitable method (see, for example, [3]). It is a class of Multi-Criteria Decision Analysis (MCA) and relies on the connection between influencing factors and hazards rather than statistics from historical databases. In this sense, the method is subjective in assigning intensities and weights of different hazards. Some examples of AHP application in multi-hazard mapping can be found in Bathrellos et al. [3], Karaman [37], and Khatakho et al. [5].

### 2.3. Vulnerability and Risk

Vulnerability lies within the characteristics or properties of the elements (structures) at risk, making them susceptible to impacts of hazards. The UNGA [24] defines vulnerability as "the conditions determined by physical, social, economic and environmental factors or processes which increase the susceptibility of an individual, a community, assets or systems to the impacts of hazards". The concept of vulnerability is used in a broad sense and with different meanings in different fields. Vulnerability is the risk determinant that is the most feasible one to manage/control/reduce through human action or interference. Vulnerability reduction is, therefore, one of the most effective forms of risk reduction. However, the quantification of vulnerability of civil engineering structures even to individual hazards, such as earthquakes, is a challenging task with many uncertainties (see, for example, [38–40]). In a multi-hazard scenario, the overall vulnerability can be different from the vulnerability to a single hazard, which makes the definition of vulnerability especially challenging. Its

complexity is due to the variations in structural material types, geometries, environments, exposure to hazards, usage, age, maintenance, and many other factors. Quantification of structural vulnerability to different types of hazards is a popular and growing research field. Structural vulnerability is mostly expressed in terms of fragility or vulnerability curves, which quantify, in a probabilistic sense, the chance of exceeding undesired states of damage conditioned to a given intensity of hazard. On a larger geographical scale, vulnerability classification of structures relies on general information about the structures, their usage, and exposure to hazards. Such classifications are commonly used for buildings. This method of vulnerability assessment was used by Nassirpour et al. [41] to rank school infrastructure in the Philippines, considering flood, wind, and earthquake hazards. A vulnerability assessment methodology for building, subjected to both single and multi-hazards, was presented by Schwarz et al. [42]. By following the principles of the European Macroseismic Scale 1998 (EMS-98, [43]), they developed vulnerability tables for different hazards (wind, flood, and earthquake). A framework to create multi-dimensional vulnerability models from vulnerability tables was also presented and applied in a few test cities in Germany. Gautam and Dong [44] present multi-hazard damage to structures in central Nepal caused by the 2015 Gorkha Earthquake and the 2017 Chhatiune Khola flash flood. A conceptual model for multi-hazard assessment of the vulnerability of historic buildings is presented by Ayala et al. [45] (2006) with an example application considering English parish churches. A comprehensive review of single and multi-hazard vulnerability and risk in historic urban areas is presented by Julia and Ferreira [46]. They also present interesting examples of the use of multi-hazard risk analysis in historic urban areas.

An interesting methodology for the risk evaluation of offshore structures subjected to ocean waves, wind, and ground motion is proposed by Bhartia and Vanmarcke [47]. They consider failure probabilities under short-term loads, as well as overall risks, due to loads of different intensities. Their results show that limit states (of failure), structural characteristics, as well as features of different types of loads interact in a complex way, controlling the relative importance of different hazards. Aggravating effects in a multi-hazard scenario are clearly demonstrated in their case-study example of ambient (ever-present wind over the sea) and seismic loads. An outline for identification of different hazards and subsequent risk assessment has been introduced by the United States Federal Emergency Management Agency, 1997, [48]. Ciurean et al. [49] present a comprehensive report of recent developments in multi-hazard processes and risks related to research, policy, and industry.

## 3. Vibration Control of Structures

Most of the structural damage caused by natural forces can be attributed to excessive vibrations. For static loads, vulnerability reduction can be achieved at the design stage by increasing stiffness and/or strength of structural elements. For existing structures, retrofitting strategies also aim to improve strength and/or stiffness of the structural elements. Similar strategies can also be used for dynamic forces such as wind and earthquakes, but newer and potentially more cost-effective solutions emerging in the scientific research are percolating to practical applications. These new solutions are not necessarily about increasing structural stiffness and/or strength. They fundamentally rely on changing the dynamic properties of the structures to make them less vulnerable to natural forces. This can, contrary to retrofitting in the traditional sense, even make the overall structure more flexible. A notable example is the well-established base-isolation technology for reducing earthquake-induced vibrations of buildings and bridges. Vibration reduction, also known as vibration control, makes use of different types of devices installed on the structure to reduce vibrations caused by different types of forces. While base-isolation, supplemental damping, and bracing systems to increase lateral stiffness and ductility have been researched and used for a long time, newer control strategies that rely mainly on dynamic devices installed on the primary structure are emerging. Depending on their mode of operation and need for external energy and/or internal feedback mechanism,

vibration control devices can be broadly classified as passive, active, semi-active, or hybrid systems. The following list gives a few examples of these different types of control systems

1.  Passive: energy dissipation, base isolation, tuned mass dampers (TMD), and tuned liquid dampers.
2.  Active: adaptive control, active bracing, and active mass damping.
3.  Semi-active: semi-active energy dissipation, semi-active isolation, and semi-active mass damping.
4.  Hybrid: hybrid bracing and hybrid mass damping.

Detailed definitions and the basic theory behind these different classes of structural control devices can be found in the seminal work of Housner et al. [50]. Most of these control concepts have been extensively investigated, and many of these systems are already installed in different types of structures.

Passive control devices are the ones most frequently used, as they don't require external energy supply. Tuned mass dampers (TMDs), fluid viscous dampers (FVD), tuned liquid dampers (TLD), and seismic base isolation (BI) are the most popular passive control systems.

An early state-of-the art review of seismically base isolated buildings is presented by Kelly [51], Buckle and Mayes [52], and Jangid and Datta [17]. They discuss different types of base isolation systems and summarize findings of the contemporary literature about their performance in seismic response control in addition to presenting a parametric study of crucial design parameters for optimal reduction in seismic performance. Patil and Reddy [19] present a state-of-the-art review of base isolation systems in seismic response mitigation. They focus on design code provisions for isolated structures and discuss effects of soft-soil and near-fault ground motions. Kunde and Jangid [18] present a state-of-the art review of seismically isolated bridges and identify some knowledge gaps in the contemporary literature. Soong and Spencer [21] discuss different types of supplemental energy dissipation systems, including passive and active dampers for structural control. They provide an informative timeline of the development of these control technologies and describe the state-of-the-art review in the context of seismic-resistant design and the retrofitting of structures. A review on the behavior of structures with passive control systems exposed to seismic loads is presented by Buckle [52]. This study discusses the advantages of passive systems in a seismic design and provides several examples of their successful applications. It also highlights limitations of passive control, considering uncertainties in seismic forces and limit states induced by unexpectedly demanding events, and points towards the need for better practical guidelines in their design and implementation.

A comprehensive review on the response control of structures by TMDs is reported by Elias and Matsagar [22]. They review different configurations of dampers, involving one or more tuned masses, installed at different locations of the structure. They report that the findings in the literature support effectiveness of TMDs in reducing wind and earthquake-induced vibrations of certain types of structures. They also identify potential obstacles, such as robustness and reliability, across different levels of loading, especially those that exceed the yield limit, causing inelastic deformations in the structure.

A state-of-the-art review of different types of structural control systems was presented by Saeed et al. [53]. Their review includes different control technologies that can be classified as active, semi-active, passive, or hybrid. They conclude that control systems have a huge potential and importance in modern structures.

Symans and Constantinou [54] present a detailed review of semi-active control systems for the seismic protection of structures and conclude that different solutions, such as stiffness control devices, electrorheological dampers, friction control dampers, fluid viscous dampers, etc., have the potential of practical feasibility in full-scale structures. Spencer and Nagarajaiah [55] also report on the state of the art of semi-active technologies for structural vibration control. They report that smart damping devices, such as Magnetorheological (MR) dampers, appear to combine desirable features of both passive and active control solutions, and they offer a viable control solution against wind and earthquake forces.

The literature on control of structures against wind or seismic forces is vast. As explained above, the state-of-the-art and recent findings, in the structural control of different kinds, have been presented in many works. As an example, one of the first comprehensive state-of-the-art reviews of structural control systems was published more than two decades ago by Housner et al. [50]. Although performance of different control schemes in a single hazard scenario, such as wind or an earthquake, is well-known and summarized in many works, structural control in multi-hazard scenario is an emerging field of research. While some interesting research has been published in this field, there is a lack of an overview of the state-of-the-art, ongoing progress, and future directions. Most of the literature in this regard is on seismic and/or wind-induced response reduction in bridges, buildings, and wind turbines. These topics are dealt with separately in the following sections.

## 4. Multi-Hazard Assessment and Control of Bridges

Bridges are lifelines of modern society. They are vulnerable to different hazards, as evidenced by several failures in the past. On 19 August 2016, a suspension railway bridge in Tolten-Chile collapsed due to train-induced vibrations [56]. On 29 August, during hurricane Katrina, the Twin Spans Bridge connecting New Orleans to Slidell, Louisiana, United States, suffered extensive damage [57]. On 21 July 2003, Kinzua Bridge in Pennsylvania, United States, was hit by a tornado with 100 mph (45 m/s) winds and collapsed [58]. On 14 January 2003, Sgt. Aubrey Cosens VC Memorial Bridge in Ontario, Canada collapsed [59] due to fatigue-induced failure of the steel hanger rods supporting the deck. On 17 January 1995, a bridge on Hanshin Expressway in Kobe, Japan collapsed during the Kobe Earthquake [60]. During the Loma Prieta earthquake in 1989, two famous bridges (Cypress Street Viaduct and San Francisco—Oakland Bay Bridge) in California, USA were heavily damaged [61,62]. The collapse of these two bridges killed forty-one persons.

Safety and reliability of bridges are controlled by a diverse set of factors related to the structural form, function, maintenance, and the hazards they are exposed to. Multi-hazard consideration is, therefore, emerging as an important topic in bridge design and safety assessment. Some of the recent advancements in this field are discussed in the following.

### 4.1. Multi-Hazard Assessment of Bridges

To understand the consequences of multi-hazard effects on the safety/reliability of bridges, a wide range of experimental and analytical studies have been conducted and reported in the literature. One of the most studied scenarios is the interaction of earthquakes with other actions: for example, traffic-load. The interaction between these hazards, when they occur concurrently, can be amplifying or diminishing (see, for example, [63,64]). Cascading effects might also be observed when bridges, partly damaged by earthquakes, are exposed to traffic (see, for example, [65,66]). Ground shaking and liquefaction induced by earthquakes can have complex interactions in bridge response, both amplifying and diminishing (see, for example, [67,68]). Another multi-hazard scenario for bridges is the simultaneous occurrence of high waves and hurricane surge (see, for example, [69–73]). Another scenario is foundation scour due to floods, which may increase the seismic vulnerability of bridges [74–81].

Aging and corrosion of bridge elements causes structural deterioration that can amplify the effect of other hazards, such as earthquakes or wind forces. The effect of deterioration caused by seismic and traffic loads on a reinforced concrete bridge is addressed by Deco and Frangopol [82], Choe et al. [83], Kumar et al. [84], Choe et al. [85], Gardoni and Rosowsky [86], Choine et al. [87], Rokneddin et al. [88], and Biondini et al. [89]. Long-span bridges are especially sensitive to wind forces, but they can also be affected by seismic excitation. A framework for the assessment of vulnerability of long-span bridges subjected to multi-hazards (seismic and wind excitation) is presented by Martin et al. [90]. A summary of recent advances in wind effects on long-span bridges, with a multi-hazard perspective, is presented in Chen et al. [91]. Studies on the multi-hazard effects and performance of bridges is summarized in Table 1.

**Table 1.** A summary of published works on bridges subjected to multi-hazard.

| Type of Hazards | Reference | Type of Bridge Structure | Aims | Main Contribution/Conclusion |
|---|---|---|---|---|
| Multi-Hazard in General | | | | |
| | Ettouney et al. [10] | Different type of bridges | Theoretical formulation | A general theory and application to structural analysis, design, life cycle costing, risk assessment, and health monitoring |
| Earthquake and Wind | | | | |
| | Martina et al. [90] | suspension bridges | Fragility analysis | Fragility surfaces for bridges exposed to multiple extreme events |
| Earthquake and Corrosion | | | | |
| | Choe et al. [85], | RC bridges | Fragility assessment | Fragility increment functions of corroding bridge columns and their application in life cycle cost and risk analysis |
| | Choe et al. [83] | RC bridges | Fragility assessment | Probabilistic models of seismic demand and fragility of corroding bridges, and their application in reliability analysis |
| | Kumar et al. [84] | RC bridges | Life-cycle cost assessment | Probabilistic model of life cycle cost considering cumulative damage, case studies highlighting dominance of cumulative seismic damage in reducing reliability |
| | Gardoni and Rosowsky [86] | RC bridges | Fragility assessment | Fragility increment functions and an example application |
| | Ghosh and Padgett [92] | Multi-span continuous highway bridges, | Fragility assessment | Time-dependent seismic fragility curves considering ageing and deterioration |
| | Simon et al. [93] | RC bridges | Fragility assessment | Losses in strength and stiffness due to corrosion have marginal effects on seismic fragilities of the case-study bridge |
| | Alipour, et al. [94] | Highway RC bridges | Fragility assessment | Time-dependent fragility models, and life cycle cost assessment |
| | Ghosh and Padgett [95] | Highway RC bridges | Loss assessment | Probabilistic framework for loss assessment using component-level cost estimates, case studies highlighting the impacts of deterioration |
| | Sung and Su [96] | RC bridge columns | Capacity, fragility, and loss estimation | Capacity models of deteriorated columns and resulting time-dependent fragility curves |
| | Zhong et al. [97] | RC bridges | Fragility assessment | Component and system level fragility curves, and a case study application |
| | Ou et al. [98] | RC bridges | Long-term performance assessment | Case studies of several bridges highlighting the need to increase design PGA to ensure adequate performance through the design life |

Table 1. *Cont.*

| Type of Hazards | Reference | Type of Bridge Structure | Aims | Main Contribution/Conclusion |
|---|---|---|---|---|
| | Rokneddin et al. [88] | RC bridges | Reliability assessment | Time-dependent fragility models of selected bridge classes, and an algorithm for reliability assessment, case studies highlight the need for accounting network-level importance in retrofit programs |
| | Akiyama and Frangopol [99] | RC bridges | Life cycle reliability assessment | Modelling spatial variability of rebar corrosion using X-ray, and a computational framework to incorporate corrosion hazard in seismic reliability |
| | Biondini et al. [89] | RC bridges | Life cycle performance assessment | Probabilistic modelling of capacities of corroding critical sections, case study highlighting undesirable effect of corrosion in seismic performance |
| | Ni et al. [100] | RC bridges | Modelling impact of corrosion on seismic performance | A new constitutive model for corroded reinforcing steel, and fragility curves at various time intervals |
| | Shekhar et al. [101] | Highway bridges | Study of realistic corrosion models and their impacts on life-cycle cost. | Framework for seismic life cycle costs from generic corrosion measures, case study demonstrating relevance of pitting versus uniform corrosion model |
| | Ghosh and Sood [102] | Highway bridges | Assessment of time-dependent capacities and more realistic degradation models in seismic fragility | A methodology for seismic fragility assessment incorporating pitting corrosion models and time-dependent capacity models, predictive equations for seismic reliability assessment over the service life |
| | Thanapol et al. [103] | RC bridges | Incorporation of spatial distribution of corrosion in seismic reliability assessment | Models to estimate steel weight loss at critical sections using spatially variable corrosion images using X-ray technology, followed by an illustrative case study |
| | Rao et al. [104] | RC bridges | Fragility assessment | Framework for seismic vulnerability assessment of deteriorating RC columns |
| | Alipour and Shafei [105] | Highway bridges | Network resilience assessment | Computational framework for risk assessment which emphasis on effect of ageing in seismic resilience |
| | Ghosh et al. [106], Rokneddin et al. [107], | Highway bridge network | Network reliability analysis | Methodology to estimate bridge fragilities using deterioration parameters from instrumented bridges in the network, and an example application |
| Earthquake and Floods | | | | |
| | Dong and Frangopol [78] | Highway bridges | Life-cycle performance assessment | A framework for time-variant loss and resilience assessment, and effect of climate change |

**Table 1.** *Cont.*

| Type of Hazards | Reference | Type of Bridge Structure | Aims | Main Contribution/Conclusion |
|---|---|---|---|---|
| | | Earthquake, Floods, and Ground-Failures | | |
| | Gehl and D'Ayala [108] | RC bridges | Fragility assessment | A Bayesian Networks based fragility surfaces |
| | | Earthquake and Scour | | |
| | Wang et al. [109] | RC bridge | Estimation of load factors | Risk-consistent load factors based on case studies |
| | Wang et al. [77] | Bridge with pile foundation | Performance assessment | Experimental evidence for effects of scour depth on failure mechanisms of piers and piles |
| | Guo et al. [110] | RC bridges | Study of the effect of time-dependent scour hazard on seismic vulnerability | Fragility surfaces and time-dependent loss estimates with two case study bridges |
| | Alipour et al. [111] | RC bridges | Investigation of scour-load modification factors in seismic assessment | Reliability-based load and resistance factors |
| | Yilmaz [112] | Highway bridges | Risk and reliability assessment | Framework for risk assessment and uncertainty analysis |
| | Chandrasekaran and Banerjee [113] | RC Bridge | Optimal retrofitting | An approach for retrofit optimization, case study results showing that column jacketing is effective |
| | Guo and Chen [114] | RC bridges | Lifecycle assessment | A framework for lifecycle assessment, case study highlighting the need for a time-sensitive assessment |
| | Han et al. [74] | High-rise pile cap foundation | Performance assessment | Seismic capacity of foundation is significantly affected by scour depth |
| | | Earthquake, Scour, and Corrosion | | |
| | Dong et al. [115] | RC bridges | Sustainability assessment | A framework for time-varying sustainability, and an illustrative application |
| | Asadi et al. [116] | RC bridge | Performance assessment | A framework for performance assessment and life cycle cost analysis |
| | | Earthquake, Scour, Corrosion, and Traffic-Load | | |
| | Deco and Frangopol [117] | Highway bridges | Risk assessment | Framework for time-varying total risk and effect of structural redundancy |
| | | Earthquake, Scour, Corrosion, Wind, Traffic-Load and Liquefaction | | |
| | Banerjee et al. [118] | Highway bridges | Review | Summary of state-of-the-art in different aspects of resilience: loss assessment, recovery actions, and maintenance |
| | | Earthquakes, Corrosion, Surge, and Wave | | |
| | Kameshwar, and Padgett [119] | RC Coastal bridges | Lifecycle risk assessment and design | Object oriented consequence-based framework for lifecycle risk assessment considering structural deterioration |

**Table 1.** *Cont.*

| Type of Hazards | Reference | Type of Bridge Structure | Aims | Main Contribution/Conclusion |
|---|---|---|---|---|
| | | Earthquakes, Wind, Tsunami, Flood, Surge, and Wave | | |
| | Gidaris et al. [120] | Highway bridges | Review | Summary of the state-of-the art in fragility and restoration models |
| | | Earthquakes, Blast and Fire | | |
| | Echevarria, et al. [121] | Concrete-filled fiber reinforced polymer tube (CFFT) bridge columns | Experimental investigation of lightly reinforced CFFT columns | Experimentally validated design equations and a formulation for displacement-based seismic design including fire protection provisions |
| | | Earthquake and traffic loads | | |
| | Sun et al. [122] | Bridges in Southeast Coastal areas of China | Study combinations of seismic and truck loads | Probabilistic methodology to combine earthquake and truck load |
| | Ghosh et al. [64] | Highway bridges | Studying effect of traffic loads on seismic reliability | Framework for joint fragility assessment, fragility surfaces, and case study application |
| | | Earthquake, Traffics-Load, and Deterioration | | |
| | Deco and Frangopol [82] | Bridges in general | Life-cycle risk assessment | Probabilistic framework for life cycle risk assessment of spatially distributed group of bridges |
| | | Earthquakes and High water | | |
| | Nikellis et al. [123] | Generic | Risk assessment | An analysis of risk metrics, stakeholder perceptions, and impact on retrofit strategies |
| | | Traffic Load and Wind | | |
| | Chen et al. [91] | Long span bridges | Review | Summary of recent advances. |
| | | Wave and Storm Surge | | |
| | Ataei, and Padgett [70] | Costal RC bridges | Capacity assessment | Probabilistic approach to global limit state capacities |
| | Ataei et al. [71] | Costal RC bridges | Fragility assessment | Development of surrogate models and uncertainty analysis |

### 4.2. Multi-Hazard Vibration Control of Bridges

Although the literature on multi-hazard vulnerability and the risk assessment of bridges is vast, retrofitting bridges for multi-hazard protection is an emerging research topic that is gaining interest. Chandrasekaran and Banerjee [113] consider three different retrofit strategies to enhance bridge performance under the multi-hazard. Wang et al. [76] note that increasing foundation stiffness can be more beneficial than increasing foundation depth in reducing seismic vulnerability of bridges subjected to scour. Sung and Su [96] use time-dependent fragility curves to estimate the total direct costs of neutralized RC bridges as a function of ground motion intensity and service time and propose it as a tool to time retrofit campaigns. Benefits of the base isolation system, as a control/retrofit solution for increasing the reliability of steel bridges subjected to ground shaking and liquefaction hazards, is demonstrated in Wang et al. [124].

To the best of our knowledge, multi-hazard considerations in vibration control of bridges has not been reported in the literature yet.

### 5. Multi-Hazard Assessment and Vibration Control of Buildings

This section provides a review of studies related to building response to multi-hazard, with emphasis on wind and seismic forces. Building response to seismic forces is controlled

by various factors, such as amplitude, duration, and frequency content of ground shaking. It also depends on the characteristics of the building itself and the underlying soil properties. Larger earthquakes produce ground motion with more energy at lower frequencies than smaller earthquakes. Large earthquakes are hazardous to all buildings, particularly to those that have natural frequencies close to the dominant frequency of ground shaking. Such a phenomenon has been observed in ground shaking and building response during past earthquakes (see, for example, [125,126]).

Wind loads contain energy at lower frequencies than seismic ground motions. Low to mid-rise buildings with relatively low vibration frequencies are therefore more susceptible to dynamic vibrations caused by seismic forces than those due to wind. Wind forces on such structures could, nevertheless, have undesired effects on components such as roofs, windows, chimneys, etc. Damage to light and improperly anchored roofs in low-rise buildings during strong wind is, therefore, of concern. In super tall buildings, wind generally induces stronger displacement response than earthquakes. Seismic loads, however, might excite higher vibration modes of such structures, resulting in high floor accelerations. This implies low inter-story drift and, therefore, lower risk of structural damage, but high floor acceleration can be critical for non-structural components [6,9,127,128]. From a structural point of view, wind loads are, therefore, critical for flexible structures, while seismic loads are more demanding on stiff structures. Occupant comfort and safety is another consideration when it comes to the response of buildings to wind and earthquake loads. Large floor accelerations can cause discomfort to occupants and may pose a safety threat due to moving objects. Floor accelerations in tall buildings are typically higher during moderate to strong ground shaking than during strong wind. Strong seismic loading is, however, typically less frequent than strong wind. From a serviceability point of view, wind action is, therefore, more critical for occupant comfort. Multi-hazard effects in buildings also need to be looked at from a life-cycle cost perspective and accumulation of damage due to multiple events: for example, wind response of a structure partially damaged by an earthquake or vice versa. Damage accumulation and fatigue due to repeated loading from frequent actions, such as moderate to strong wind, is also an important consideration.

*5.1. Multi-Hazard Assessment of Buildings*

Huang [129] provides a comprehensive account of the dynamic responses of high-rise buildings under multiple hazards. It presents performance assessment methods and case study investigations using high-rise buildings in Hong Kong. Various factors, such as seismic source-to-site distance, recurrence periods, ground shaking amplitude, building height, damping ratios, properties of wind forces, etc., were considered in the analysis. The results show that seismic loads result in a higher floor acceleration response and lateral forces but weaker torsional forces and a lower displacement response compared to wind forces. The height of the buildings was also found to be an important parameter, with wind response being more sensitive to variation in height than seismic response. The results also showed that wind response is more strongly influenced by the level of damping of the building than seismic response.

Chen [127], and Rasigha and Neeladharan [128] report differences in the seismic and wind responses of mid-rise to high-rise buildings. Aly [6], as well as Aly and Abburu [7] present the responses of tall buildings subjected to wind and seismic forces. In these assessments, two tall buildings (76-story and 54-story) were considered for finite element analysis. They found that ground motions excite higher vibration modes in buildings, resulting in lower inter-story drift than wind forces, but higher floor accelerations last for a shorter time. Wind actions are, therefore, critical from an occupant comfort and serviceability consideration. Tall structures designed for strong wind may possess an adequate capacity against moderate ground shaking, but they might suffer non-structural losses due to high floor accelerations. A framework for life-cycle loss estimation, of non-structural damage in tall buildings under wind and seismic loads, is presented by Venanzi et al. [9]. Their framework assumes that damaged structures are restored to their

original condition after each hazardous event. Hazardous events are not simultaneous, and small maintenance costs are neglected. Their results show that for drift-dependent damages, wind forces are costlier than seismic forces. Seismic forces are costlier, in terms of non-structural damage, due to high floor accelerations. These observations are consistent with results reported in other studies, [6,7]. Antoun [130] studied the performance of a 74-story building located in Miami to evaluate the expected losses associated with a multi-hazard (wind and earthquake forces). Performance-based approaches were used for earthquake, wind, and hurricane forces. Monetary losses corresponding to structural and non-structural damage, as well as occupant discomfort, was estimated. They report that losses due to façade damage are dominant for high probabilities of exceedance, whereas structural damage becomes dominant at lower probabilities of exceedance.

Zhang et al. [131] proposed a framework for the damage risk assessment of high-rise buildings exposed to wind and seismic forces acting separately and concurrently. They used recorded earthquake and wind data, over a period of about 47 years, to estimate hazard curves for wind and seismic forces as well as copula-based bi-hazard surfaces. They then performed multi-hazard fragility assessment and estimated damage probabilities for separate and concurrent hazard models. Their results show that damage probability due to bi-hazards dominates the total damage probability in most damage states. They highlight the need for multi-hazard considerations in the design and evaluation of tall structures subjected to wind and seismic forces. Damage risk assessment and cost-benefit analysis of mitigation strategies, in residential buildings subjected to hurricane and seismic forces, are discussed in Li [132], giving a comprehensive overview of factors that are important in risk assessment, as well as their roles and impacts in hazard mitigation. The risk-cost-benefit framework, based on life-cycle and scenario-case analyses presented by Li [132], incorporate probabilistic modelling of hazards, structural fragility, and expected costs during different service intervals.

Multi-hazard consideration in performance-based engineering and performance-based design criteria, addressing wind and seismic forces, has been researched extensively in the literature. Chiu and Chock [133] present one of the first applications of the performance-based engineering approach in a multi-hazard scenario.

A probabilistic framework, for the multi-hazard risk assessment of reinforced concrete buildings subjected to seismic and blast loads, is discussed in Asprone et al. [134]. Annual risk of structural collapse, considering seismic action and progressive collapse due to blast forces, is formulated in this study. They conclude that the Monte Carlo (MC) simulation is suitable for calculating probability of progressive collapse, as well as for identifying critical blast scenarios.

Multi-hazard performance of different structural elements, such as columns, frames, plates, walls, etc., have been reported by many researchers. Resistance capacity of precast segmental columns, subjected to impact and cyclic loading, is investigated experimentally by Zhang et al. [135]. They found that, compared to monolithic columns, segmental columns (precast segments joined together, often with pre-stressed tendons) possess better ductility and sustain lower residual drift under cyclic loading. Under impact loading, segmental columns were found to have better self-centering capacity. They showed that shear resistance of such columns can be significantly improved by introducing concrete shear keys, but it comes at some cost related to stress-concentration and potential damage to concrete segments.

Rachel [136] presents a methodology for the resilience assessment of buildings subjected to seismic, wind, fire, and various post-earthquake scenarios. The results of this study showed that post-earthquake fire resilience in moment frame buildings is independent of seismic damage if frame connections are intact. The results also showed that multi-hazard resilience of moment resisting frame buildings can be improved by strengthening and/or fire-proofing gravity columns. Shin [137] presents multi-hazard performance evaluation matrices for retrofitted non-ductile reinforced concrete buildings subjected to seismic and blast loads.

Unnikrishnan and Barbato [138] investigated multi-hazard interaction on the performance of low-rise wood-frame buildings. Chulahawat and Mahmoud [139] present an algorithm to optimize building systems, with suspended floor slabs subjected to wind and seismic hazards, and observe that tall buildings with such systems are effectively optimized for both wind and seismic forces without a significant trade-off on performance to individual hazards.

### 5.2. Multi-Hazard Vibration Control of Buildings

Vibration control of buildings subjected to wind or seismic forces has been extensively researched. Vibration control of buildings in multi-hazard scenarios is, on the other hand, not as extensively studied. Some important studies in this area are summarized in Table 2. Performance assessment of control devices, their optimization, and life-cycle cost analysis are the main issues that have been addressed in these studies. Wind and earthquake forces are the most considered hazard in these studies. Most of these studies present traditional control systems such as passive TMDs, passive energy dissipation devices, viscous fluid dampers, multiple tuned passive TMDs, etc. Some recent advances in this area include inerter-based TMDs (Djerouni et al., [140]; Djerouni et al. [141]; Djerouni et al. [142]; Marian and Giaralis [143]), glass curtain wall TMDs (Bedon and Amadio [144]), and sliding floor isolators (Chulahwat and Mahmoud [139]; Mahmoud and Chulahwat [145]).

**Table 2.** A summary of published works on vibration control of the building subjected to multi-hazard.

| Reference | Hazards | Structure | Control System | Main Contribution/Conclusion |
|---|---|---|---|---|
| Cao et al. [146] | Wind, Blast and Earthquake. | 5- and 39-story benchmark buildings | Semi-active friction damper | A new controller called input space dependent controller (ISDC) which is more effective than sliding mode controller. |
| Mahmoud and Chulahwat [145] and Chulahqat and Mahmoud [139] | Wind and Earthquake | 7- and 10-story steel frame buildings. | Sliding floor isolation. | A modified covariance matrix adaptation evolution strategy (CMA-ES) algorithm |
| Dogruel and Dargush [147] | Wind and Earthquake | 16-story steel frame building. | Passive Energy Dissipation (PED). | Methodology for optimal life-cycle cost estimation, and optimal design of retrofitting |
| Shalom et al. [148] | Wind and Earthquake | 76-storey building. | Multiple Tuned Mass Dampers (MTMDs) | Life-cycle cost-based optimization framework |
| Roy and Matsagar [149,150] | Wind and Earthquake | 9-, 20-, and 25-story steel frame buildings. | PED | Optimal retrofits for earthquakes result in undesirable effects on wind response, and vice versa; damper performance is sensitive to site-specific hazard |
| Chapain and Aly [151] | Wind and Earthquake | 76-story building | Viscous Fluid Dampers (VFDs) | VFDs are effective in multi-hazard control |
| Elias and Matsagar [152] | Wind and Earthquake | 76-story and 20-story building | TMD | Optimally placed and tuned TMDs are effective in multi-hazard control |
| Elias et al. [153] | Wind and Earthquake | 76-story building | MTMDs | MTMDs with equal stiffness are better than those with equal masses. |
| Bedon and Amadio [144] | Earthquake and blast | 4-storey steel frame building | Glass curtain walls as passive TMD | Glass curtain walls can be utilized as distributed TMD for vibration mitigation |
| Gong [154] | Wind and Earthquake | 5-,9-, and 20-story buildings | Variable Friction Cladding Connection (VFCC) | Experimental and analytical evidence demonstrating effectiveness of VFCC. |

## 6. Multi-Hazard Assessment and Control of Wind Turbines

The tall and slender geometry of wind turbine towers and the large top mass of the turbine and the rotors make wind turbines sensitive to both wind and seismic excitation.

Wind and seismic loading have been the two most common environmental actions considered for research on the performance assessment of wind turbine towers. For offshore turbines, wave loading is also an important factor.

### 6.1. Multi-Hazard Assessment of Wind Turbines

Maryam [155], as well as Maryam and Gardoni [156] highlight the importance of multi-hazard consideration in site-selection and design of wind turbines. They present a multi-hazard probabilistic framework to evaluate the structural reliability of offshore wind turbines. Considering wind and seismic action, their results show that annual probabilities of failure are higher when seismic action is considered. Comparing two identical wind turbines, one in the Gulf of Mexico and the other off the coast of California, they conclude that, although the latter location is more favorable in terms of power production, annual probabilities of failure are higher due to higher seismicity. Avossa et al. [157] present a Monte Carlo simulation-based framework for the estimation of multi-hazard fragility curves of wind turbine structures. They provide an example application of the framework, to derive failure probabilities of a prototype wind turbine structure, conditioned on wind velocity and peak ground acceleration for different operational states of the turbine. Their results show that aerodynamic damping plays an important role in the seismic fragility. Fragility in an operational state, for seismic action in the fore-aft direction, increases with wind speed up to the rated wind speed, after which it starts to decrease. When the rotor is operating at the rated condition or is parked, the probability of failure is larger than 50% and the peak ground acceleration exceeds about 70% of the acceleration of gravity. Campo and Estrada [158] present similar conclusions regarding the importance of aerodynamic damping, stating that, while wind action is more damaging at the operational state, seismic action can be more threatening when the rotor is parked. Katsanos et al. [159] report, for a 5 MW offshore wind turbine, that seismic action contributes more than wind and wave action to structural demands such as base moment and tower-top displacement. They also report on the fragility of sensitive equipment located in the nacelle, which are found to be prone to severe damage at moderate ground shaking intensity. Zuo et al. [160] investigated the fragility of a prototype 5 MW offshore wind turbine structure subjected to aerodynamic forces and wave loading. They considered different operational states of the rotor and derived fragility curves for both the supporting tower and the rotor blades. Their results show that, when the wind speed is between the cut-in and cut-out range, exceedance probability of the blade failure is much higher than that of the tower failure. They also highlight the impact of aerodynamic damping in reducing wind-induced vibrations of the tower. Zuo et al. [161] studied the effect of soil structure interaction (SSI) on the 5 MW offshore prototype model. Their results show that the fore-aft displacement demand on the tower is significantly affected by SSI. Asareh et al. [162] investigated the fragility of a 5 MW wind turbine prototype subjected to wind and seismic action. Their results show that failure due to exceedance of tower-tip displacement and rotation is more likely than yielding or buckling of the tower.

### 6.2. Multi-Hazard Vibration Control of Wind Turbines

Vibration control of wind turbine structures, subjected to the combined actions of wind, waves, and earthquake ground motions, is extensively reported in the literature. These studies are mostly aimed at the optimization and performance assessment of control systems. A summary of relevant studies on the vibration control of wind turbine structures subjected to multi-hazard is presented in Table 3.

**Table 3.** A summary of published works on vibration control of wind turbine structures subjected to multi-hazard.

| Reference | Hazard | Structure | Control System | Main Contribution/Conclusion |
|---|---|---|---|---|
| Xie et al. [163] | Wind and Waves | Offshore, 5 MW, barge-type floating | TMD on the platform | Modelling of drivetrain dynamics, optimization of TMD parameters, simulation results confirm the importance of drivetrain dynamics and that TMDs are effective in vibration suppression |
| He et al. [164] | Wind and Waves | Offshore, 5 MW, barge-type floating | TMD in the nacelle | TMDs are effective in reducing standard deviation of tower-top displacements, with upto 50% reduction for TMD mass ratio of 2% |
| Stewart and Lackner [165] | Wind and Waves | Offshore, 5 MW, monopile | TMD in the nacelle | Misalignment of wind and wave load significantly increases base moment in the side-side direction, which can be reduced by 40% with a TMD |
| Zhao et al. [166] | Wind, wave and seismic | Offshore, scaled model, monopile | TMD in the nacelle | Shake table tests for modal identification and estimation of aerodynamic damping, results show that TMDs are effective in reducing seismic response, effectiveness increases with rotation speed of blades |
| Sun and Jahangiri [167] | Misaligned wind-wave and seismic | Offshore, 5 MW, monopile | Pendulum tuned TMD(PTMD) in the nacelle | PTMDs are slightly more effective than two linear TMDs with equivalent mass and their stroke is smaller |
| Sun and Jahangiri [168] | Misaligned wind-wave | Offshore, 5 MW, monopile | PTMD in the nacelle | Increase in fatigue life due to PTMD is 50% higher than that due to dual linear TMDs. |
| Sun et al. [169] | Misaligned wind-wave and Seismic load | Offshore, 5 MW, monopile | PTMD in the nacelle | PTMD is more robust than dual linear TMDs, and with a mass ratio of 2% reduction in short-term fatigue damage is reduced by up to 90%. |
| Hu et al. [170] | Wind and waves | Offshore, 5 MW, barge-type floating | Tune Mass Damper Inerter (TMDI) in the nacelle | TMDI are more effective than TMD but there is a trade-off between fore-aft load control and device stroke; performance superior to TMD can be achieved for comparable device stroke |
| Zuo et al. [171] | Wind and Waves | Offshore, 5 MW, monopile | MTMD | MTMDs are efficient and robust in reducing the out-of-plane vibration of blades and the tower in parked and operational conditions. |
| Zuo et al. [172] | Wind, Waves and Earthquake | Offshore, 5 MW, monopile | MTMD | Multi-mode control using MTMDs are more efficient than STMDs in multi-hazard scenarios |
| Hussan et al. [173] | Wind, Waves and Earthquake | Offshore 5 MW with standard jacket foundation | MTMD | SSI plays important role in MTMD performance, often over-estimates it |
| Altay et al. [174] | Wind and Earthquake | Onshore 5 MW | TMD and Tuned Liquid Column Damper TLCD | Resonant tower vibrations at lower wind speeds are effectively reduced by TMDs and TLCDs, transient tower vibrations at higher wind speeds are less effectively reduced, only nominal control of seismic-induced vibrations |
| Colwell and Basu [175] | Wind and Waves | Offshore, monopile | TLCD | TLCDs are effective in reducing peak response and increasing fatigue life |
| Dezvareh et al. [176] | Wind and Waves | Offshore, 5 MW, jacket type | Tuned Liquid Column Gas Dampers (TLCGD) | Effective in reducing nacelle displacement and acceleration protecting the tower structure and acceleration-sensitive nacelle devices |

**Table 3.** *Cont.*

| Reference | Hazard | Structure | Control System | Main Contribution/Conclusion |
|---|---|---|---|---|
| Bargi et al. [177] | Wind, Waves and Earthquake | Offshore, 5 MW, jacket tupe | Tuned Liquid Column Gas Dampers (TLCGD) | Nacelle acceleration is better controlled under wind-wave excitation while nacelle displacement is better controlled under seismic load, heavier devices are more efficient but less robust against detuning |
| Sun [178] | Wind, Waves and Earthquake | Offshore, 5 MW, monopile | Semi-active TMD (S-TMD) | Semi-active TMDs are more efficient than passive TMDs |
| Hemmati and Oterkus [179] | Wind, Waves and Earthquake | Offshore, 5 MW, monopile | S-TMD | S-TMD provide better control than passive TMDs with as much as 4 times lower mass |
| Rezaee and Aly [180] | Wind, Wave, Earthquake, | Land based and offshore, 5 MW | MR damper-used as S-TMD and with outer bracing | The dampers are efficient in reducing strong vibrations and its duration during seismic loading. While displacement control starts early during ground shaking, acceleration control lags behind by a few seconds |
| Xie and Aly [181] | Wind and Earthquake | Various | TMD, TLD, VD, A-TMD, S-TMD, and TLCD, | A state-of-the-art review for evaluating the performance of the various types of control systems |
| Rezaee and Aly [182] | Wind and Waves | On-land 5 MW | TMD, TLCD, VD, and tuned sloshing damper (TSD) | Comparative study of different dampers show that VDs are the most robust, and that TSDs are effective at a wider range of frequencies |
| Zhao et al. [183] | Wind and Earthquake | On-land, 1.5 MW | Scissor-Jack Braced Viscous Damper (VD-SJB) | VD-SJB is effective and practical in reducing vibrations, seismic-vibration reduction in fore-aft direction is lower in operating condition than in parked condition |
| Zuo et al. [184] | Wind, Waves and Earthquake | Various | Various | State-of-the-art review |
| Rahman et al. [185] | Wind, Waves and Earthquake | Various | Various | Literature review |

## 7. Concluding Remarks

This work is an attempt to summarize a vast body of research literature on multi-hazard effects on structures and their vibration mitigation measures. Aspects such as performance assessment, fragility modelling, life-cycle cost assessment, and vibration control in a multi-hazard scenario are covered. The main emphasis is on wind and seismic actions on major infrastructure, such as bridges, buildings, and wind turbine towers. Understanding of multi-hazard scenarios in a probabilistic sense and mapping them out for engineering design is an evolving field. At a local scale, multi-hazard mapping using the Analytical Hierarchy Process (AHP) is gaining popularity. Multi-hazard mapping at regional scales remains a challenging task, demanding more research on unifying frameworks that standardize and unify existing probabilistic hazard assessment methods used for different natural actions. Some recent advances in multi-hazard vulnerability of buildings include multi-dimensional vulnerability modelling. Fragility modelling in a multi-hazard scenario is still a growing field of knowledge, with many unresolved questions. Some examples of such unresolved issues relate to: (i) definition of intensity measures of hazards that might interact with each other, resulting in overall effects that are of different nature than those due to individual hazards; (ii) definition of joint probabilities of exceedances of intensities of different types of hazards; (iii) lack of empirical data on actual damage recorded in multi-hazard scenario, etc.

Multi-hazard assessment of bridges is a widely studied topic. Most studies in this area focus on seismic loads and corrosion. Other effects, such as wind loads, scour, traffic loads, etc., in conjunction with seismic loads, have also been reported. Published literature on

bridges subjected to seismic loads and corrosion focus on the fragility assessment. Multi-hazard effects in such assessments are generally modelled through fragility increment functions, damage accumulation, and time-dependent fragility curves. Probabilistic load and resistance factors, for different hazards affecting bridges, is another widely reported research theme. In most cases, such fragility models are intended for a life-cycle cost and risk analysis. While multi-hazard fragility of individual bridges is widely reported, there are only a few deals with bridge networks. More research is needed in capacity modelling, risk metrics, stakeholder perceptions, and load combinations.

The literature on multi-hazard effects on buildings is dominated by wind and earthquake loads. Performance-based engineering frameworks, progressive damage and collapse modelling, resilience assessment, multi-hazard performance evaluation metrics, etc., are some of the recent advances in damage risk and life cycle cost analysis of buildings. A variety of control systems such as passive TMDs, passive energy dissipation devices, viscous fluid dampers, multiple tuned passive TMDs, have been investigated in control of buildings subjected to wind and seismic forces. Some recent advances in this area seem promising and practically appealing: for example, sliding floor isolators, glass curtain wall TMDs, and variable friction cladding connections [VFCC]. While the literature on the vibration control of buildings subjected to wind or seismic action is vast, relatively few studies have addressed their simultaneous occurrence. More research is needed on probabilistic treatment of multi-hazard load cases, robustness of control devices against uncertainties in structural properties, as well as loading the feasibility of control from a life-cycle perspective.

Seismic and wind forces are the two most considered environmental actions in the performance assessment and vibration control of wind turbine structures. Wave action, and effects of wave/wind misalignment in offshore wind turbines is also widely researched. Multi-hazard probabilistic framework for reliability assessment of offshore wind turbines is relatively well-established. Monte Carlo simulation-based frameworks for multi-hazard fragility assessment are recently emerging. For land-based wind turbines, several studies have highlighted the role of aerodynamic damping in response to combined action of wind and earthquakes. Most of the published work on vibration control of wind turbines focuses on structural fragility of the supporting tower. Offshore wind turbines subjected to wind and waves is the most investigated scenario. The most reported control device is passive TMD placed on the nacelle, although use of TMD on the platform of a barge-type floating turbine has also been reported to be effective. Most of the studies conclude that control devices are effective in reducing multi-hazard fragility. Recent advances in multi-hazard control of wind turbines include interesting innovations such as braced viscous dampers, and semi-active control systems. Effect of ground motion variability on control performance is an area that needs to be studied better. Control performance against impulsive loads caused by, for example, near-fault ground motions (see, for example, Rupakhety et al. [186]; Elias et al. [187]; Sigurðsson et al. [188], Jami et al. [189]) also need to be investigated better. In addition, fragility of rotor blades and effects of drivetrain dynamics need more attention.

In most of structural vibration control studies reported in the literature, the structure is assumed to remain elastic, which may not be realistic in extreme loading conditions. Inelastic deformations of the structure can result in de-tuning of the control device resulting in lower performance. Control optimization and performance assessment of yielding structures subjected to multi-hazard scenarios, as well as damage accumulation due to multiple hazards occurring over the useful life of a structure, need to be investigated and better understood to facilitate practical applications of control systems in actual engineering projects.

**Author Contributions:** M.J.: Conceptualization, research, literature study, writing, and review; R.R.: methodology, writing, and review; S.E.: review; B.B.: writing, and review; J.T.S.: writing and review. All authors have read and agreed to the published version of the manuscript.

**Funding:** Matin Jami is supported by a doctoral grant from the University of Iceland. RR acknowledges support from the University of Iceland Research Fund.

**Institutional Review Board Statement:** Not applicable.

**Informed Consent Statement:** Not applicable.

**Data Availability Statement:** Not applicable.

**Conflicts of Interest:** The authors declare no conflict of interest.

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
