# Peer review of "Recent Advancement in Assessment and Control of Structures under Multi-Hazard"

_applsci, doi:10.3390/app12105118_

Round 1

Reviewer 1 Report

  • There seems to be a need for a figure that illustrates the overall process of the article. Because the items are segmented and explained in different parts, the need for a figure for making an easier understanding of the writing process.
  • In the reviews sections of other authors' articles, a series of articles are reviewed in each section, but we do not know what part of the total articles are available and whether other articles are available in these sections or not. Therefore, it is necessary to clarify and bring numbers to prove that.
  • It is also important to indicate in what keywords these articles were accessed and from what source they were used to find the articles. 

Author Response

We thank the reviewer for constructive comments which helped improve our manuscript. Response to the reviewer comments and changes made in the manuscript to address them are provided below. In the revised manuscript, changes of minor editorial nature are highlighted in cyan, while new additions are highlighted in yellow.

1. There seems to be a need for a figure that illustrates the overall process of the article. Because the items are segmented and explained in different parts, the need for a figure for making an easier understanding of the writing process.

Thank you for the comment. We have now described the methodology used in developing the thematic structure of the paper and added two new figures (Fig 2 and Fig 3) in the revised manuscript.

2. In the reviews sections of other authors' articles, a series of articles are reviewed in each section, but we do not know what part of the total articles are available and whether other articles are available in these sections or not. Therefore, it is necessary to clarify and bring numbers to prove that.

Thank you for your comment. The introduction section of the paper has been expanded with a description of how the articles were searched for and classified into different sections. All the articles listed are available.

3. It is also important to indicate in what keywords these articles were accessed and from what source they were used to find the articles. 

Please refer to our response to your comment number 1and 2 above.

Reviewer 2 Report

The reviews presents a very relevant review of literature. Please incorporate the following suggestions

  1. In the introduction section, authors should structure it in such a way that it provides literature background on earthquake - magnitude, effect on structures, associated multi-hazards, its combined effect on structure and mitigation measures. Same way, should discuss the effect of wind on structures based on wind velocity and its classification
  2. Section 2 is a general discussion on risk - vulnerability and hazard. A small introduction in general should be followed by specific discussion on earthquake and storms (I persume that authors intend to discuss on wind intensity equivalent to storms), their impact on structures , risk assessment for these specific cases.
  3. Section 2 does not provide adequate information on risk assessment of earthquakes and wind related disasters/hazards
  4. p.8, line 367 - chapters can be replaced with sections 
  5. Section  4 and 5 can include other life line structures like pipe lines in case of earthquakes
  6. Section 4 and 5 must be restructured for better readability
  7. The discussion section must bring out the methods adopted for assessing damage of structures and suitable mitigation measures along with their merits and limitations
  8. Include a conclusion to the manuscript

Author Response

The reviews presents a very relevant review of literature. Please incorporate the following suggestions.

Thank you very much for your valuable time and efforts in reviewing our manuscript. Our response to each of your comments/suggestions are outlined below. In the revised manuscript, changes of minor editorial nature are highlighted in cyan, while new additions are highlighted in yellow.

1. In the introduction section, authors should structure it in such a way that it provides literature background on earthquake - magnitude, effect on structures, associated multi-hazards, its combined effect on structure and mitigation measures. Same way, should discuss the effect of wind on structures based on wind velocity and its classification.

We appreciate your comment. Our main focus on this article is on multi-hazard effects followed by the secondary focus of vibration control. Although it is correct that earthquakes and wind forces are the most discussed phenomenon in this work, other hazards are also considered, but mostly in a multi-hazard scenario. The introduction section is therefore aimed at “setting the scene” for emphasizing the importance of multi-hazard consideration rather than explaining the effects of individual hazards like earthquakes and wind on structures. Such an endeavor is perhaps worthy but would require a lot more work considering the vast literature in this area. It is therefore our intention to keep this section short and focused on the discussion of multi-hazard and its relevance to structures in a broad sense. To clarify this, we have described in the Introduction section, the methodology used in developing the thematic structure of the paper and added two new figures (Fig 2 and Fig 3) in the revised manuscript.

2. Section 2 is a general discussion on risk - vulnerability and hazard. A small introduction in general should be followed by specific discussion on earthquake and storms (I persume that authors intend to discuss on wind intensity equivalent to storms), their impact on structures, risk assessment for these specific cases.

Please refer to our response to your comment 1 above. Please also note that elements such as hazard, vulnerability, and risk in this section are being discussed from the multi-hazard perspective only. Detailed treatment of individual hazards such as earthquakes and winds would require a lot more work and space and are outside the scope of this work.

3. Section 2 does not provide adequate information on risk assessment of earthquakes and wind related disasters/hazards

We agree with the reviewer. This is not the objective of this work and a vast literature, much larger than that reviewed in this work, exists in this area. Therefore, we focus on multi-hazard related studies only. Please also refer to our response to your comment 1 above.

4. p.8, line 367 - chapters can be replaced with sections 

We agree and have made corresponding changes in the revised manuscript.

5. Section  4 and 5 can include other life line structures like pipe lines in case of earthquakes

We agree, but there are very few studies addressing vibration control of pipe lines or its multi-hazard risk assessment. Some structures like transmission systems have been addressed in the literature to some extent, but we did not find sufficient studies to make a meaningful review in the context of vibration control and /or assessment in the multi-hazard scenario. We clarify this in the revised Introduction, where the process of searching, filtering, and classifying the reviewed articles is described in more detail.

6. Section 4 and 5 must be restructured for better readability

Thank you for the comment. In the revised manuscript, we have made the structure of section 4, 5, and 6 similar. They each contain sub-sections (i) multi-hazard scenario (ii) vibration control. Since the second section regarding bridges is not adequately addressed in the literature, it is rather short, while the first section is vast, and is summarized in a tabular format. For buildings and wind turbines, vibration control studies in multi-hazard studies are numerous, therefore, they are summarized in a tabular form, while multi-hazard scenarios are discussed in text/descriptive format.

7. The discussion section must bring out the methods adopted for assessing damage of structures and suitable mitigation measures along with their merits and limitations

We appreciate the comment. Our focus is not on damage assessment but rather performance assessment and vibration control. Therefore, we discuss recent advancements and lack of knowledge in these areas in this section rather than comparing different available solutions. In the revised manuscript, we have shortened this section and renamed it as “Concluding Remarks”.  

8. Include a conclusion to the manuscript

The Discussion section has been shortened and renamed as “Concluding Remarks”

Reviewer 3 Report

This is a very useful contribution to the problem of multi-hazard effects on structures and their control. In fact, such "reviews of review papers" may play the role of modern, meta-scientific research analyses which is not the case in this submission but not far from this.  In the opinion of this reviewer, it is so because the conclusions of this paper could still be deeper and more specific.  Below are some examples - yet it is advised that the Authors could further extend their critical, meta-research approach of this study.

lines 618 - 630:  try to be more specific, in particular in lines 628 - 630: What questions? Can you bring particular questions?

lines 645 - 647: This conclusion is rather trivial. The Authors should write much more about this. For example, please try to conclude for which type of risk massive roofs of single-story buildings are favorable and for which are unfavorable?

In chapter 7th (lines 604-721) try to address similar SPECIFIC problems of optimum design or other specific problems of multi-hazard  - as concluded from so many references quoted in this submission.

Author Response

This is a very useful contribution to the problem of multi-hazard effects on structures and their control. In fact, such "reviews of review papers" may play the role of modern, meta-scientific research analyses which is not the case in this submission but not far from this.  In the opinion of this reviewer, it is so because the conclusions of this paper could still be deeper and more specific.  Below are some examples - yet it is advised that the Authors could further extend their critical, meta-research approach of this study.

Thank you for your positive and constructive feedback. We appreciate your efforts in helping us improve the manuscript. We fully agree with you that meta-scientific research is very valuable, and also that the presented study is not quite that. The reason for this is that the work was not planned and structured as a meta-study of the literature but emerged out of the literature search of the first author during his doctoral study. The aim was to fully understand the literature in multi-hazard control of structures. The main methodology used to search for articles, filter them, and classify them in a thematic manner is described in the revised version of the manuscript. Nevertheless, the present work cannot be considered a systematic meta-scientific research, but an effort to provide a summary of what is known or unknown in the area being studied. However, since a large number of papers have been acquired, studied, tagged, and classified, there is a possibility to expand the study further and conduct a well-structured meta-analysis. We thank the reviewer for this suggestion, and reserve this for a future endeavor.

Our responses to your comments are listed below. In the revised manuscript, minor changes of editorial nature are highlighted in cyan, while new additions or major changes are highlighted in yellow.

1. lines 618 - 630:  try to be more specific, in particular in lines 628 - 630: What questions? Can you bring particular questions?

We have provided some examples in the revised manuscript by adding the following text (line 692-698)

Fragility modelling in a multi-hazard scenario is still a growing field of knowledge, with many unresolved questions. Some examples of such unresolved issues relate to (i) definition of intensity measures of hazards that might interact with each other and resulting in overall effects that are of different nature than those due to individual hazard (ii) definition of joint probabilities of exceedances of intensities of different types of hazards (iii) lack of empirical data on actual damage recorded in multi-hazard scenario, etc.   

2. lines 645 - 647: This conclusion is rather trivial. The Authors should write much more about this. For example, please try to conclude for which type of risk massive roofs of single-story buildings are favorable and for which are unfavorable?

Lines 645-647 in the original manuscript deals with bridges rather than buildings, so we assume the reviewer is referring to the paragraph following these lines. In the revised manuscript, we have shortened the Concluding Remarks section. Your suggestion is incorporated in Section 5 of the revised manuscript. The text in the revised manuscript reads (Line 501-524)

Wind loads contain energy at lower frequencies than seismic ground motions. Low to mid-rise buildings with relatively low vibration frequency are therefore more susceptible to dynamic vibrations caused by seismic forces than those due to wind. Wind forces on such structures could, nevertheless, have undesired effects on components such as roofs, windows, chimneys, etc. Damage to light and improperly anchored roofs in low-rise buildings during strong wind is therefore of concern. In super tall buildings, wind generally induces stronger displacement response than earthquakes. Seismic loads, however, might excite higher vibration modes of such structures, resulting in high floor accelerations. This implies low inter-story drift and therefore lower risk of structural damage, but high floor acceleration which can be critical for non-structural components (Venanzi et al., 2018; Chen, 2012; Rasigha and Neeladharan, 2016; and Aly and Abburu, 2015). From a structural point of view, wind loads are therefore critical for flexible structures, while seismic loads are more demanding on stiff structures. Occupant comfort and safety is another consideration when it comes to response of buildings to wind and earthquake loads. Large floor accelerations can cause discomfort to occupants and may pose safety threat due to moving objects. Floor accelerations in tall buildings are typically higher during moderate to strong ground shaking than that during strong wind. Strong seismic loading is, however, typically less frequent than strong wind. From a serviceability point of view, wind action is therefore more critical for occupant comfort. Multi-hazard effects in buildings also need to be looked at from life-cycle cost perspective and accumulation of damage due to multiple events, for example, wind response of a structure partially damaged by an earthquake or vice versa. Damage accumulation and fatigue due to repeated loading from frequent actions such as moderate to strong wind is also an important consideration.

3. In chapter 7th (lines 604-721) try to address similar SPECIFIC problems of optimum design or other specific problems of multi-hazard  - as concluded from so many references quoted in this submission.

We appreciate the feedback. The objective of this study is to summarize recent advances in this area rather than specific problems of optimization in multi-hazard consideration. We therefore keep this section broad rather than specific with the theme of first summarizing new developments and then indicating gaps in knowledge. More specific issues as those suggested by the reviewer are summarized in different tables in Sections 4-6. Reproducing these observations in text in the Concluding Remarks sections would make the article too long.

Reviewer 4 Report

The article has a clear objective. The authors have achieved their goal sufficiently. The article is an important contribution to engineering.  A summary of the knowledge in the area is presented with emphasis on the most important research results. Both in terms of time and scope, this task was challenging for the authors.

Author Response

The article has a clear objective. The authors have achieved their goal sufficiently. The article is an important contribution to engineering.  A summary of the knowledge in the area is presented with emphasis on the most important research results. Both in terms of time and scope, this task was challenging for the authors.

Thank you for your time and effort in reviewing the manuscript. We really appreciate your positive feedback on our work.

Round 2

Reviewer 1 Report

The changes are fine and the manuscript is ready to publish